# Assessment of Glucocorticoid Removal by UVA/Chlorination and Ozonation: Performance Comparison in Kinetics, Degradation Pathway, and Toxicity



Ai Zhang [1], Xinyuan Jiang [1], Qiancheng Wang [1], Siyu Hao [1], Dahai Zhu [1], Jie Wang [2,*], Ce Wang [3] and Mingyan Liu [4]

[1]   College of Environmental Science and Engineering, Donghua University, 2999 North Renmin Road, Shanghai 201620, China
[2]   Fishery Machinery and Instrument Research Institute of Chinese Academy of Fishery Sciences, 63 Chifeng Road, Shanghai 200092, China
[3]   Shanghai Zhuoyuan Water-Ecological Environmental Engineering Co., Ltd., Shanghai 200003, China
[4]   China Tiegong Investment & Construction Group Co., Ltd., Beijing 101399, China
*   Correspondence: wangjie4545@126.com; Tel./Fax: +86-021-67792538

**Abstract:** Glucocorticoids (GCs) have drawn great concern due to widespread contamination in the environment and application in treating COVID-19. This work aimed to compare the performance of UVA/chlorination and ozonation on GC removal in terms of removal efficiency, degradation pathway, and toxicity change, with fluocinolone acetonide (FA), triamcinolone acetonide (TA), and clobetasol propionate (CP) as target compounds. The results showed that both UVA/chlorination and ozonation could degrade GCs. Compared with UVA/chlorination (removal efficiency of 89% for FA, 86% for TA, and 90% for CP at 7 h), ozonation (removal efficiency of 90% for FA, 96% for TA, and 98% for CP at 15 min) was more effective in GC removal. Photodegradation contributed most to GC removal during UVA/chlorination, while $O_3$ molecules were the main functional species during ozonation. H-abstraction, dechlorination, carbon–carbon bond cleavage, and ester hydrolysis were proposed for both UVA/chlorination and ozonation based on the identification of intermediates. However, ozone tended to attack C=C double bonds, resulting in the cracked benzene ring of GCs, while chlorine was more likely to attack alcohol and ketone groups. Although most GCs were removed during ozonation and UVA/chlorination, their acute toxicities slightly declined. Compared with UVA/chlorination, ozonation was more effective in toxicity reduction.

**Keywords:** glucocorticoids; ozonation; UVA/chlorination; degradation intermediates; acute toxicity

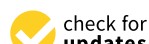



## 1. Introduction

Glucocorticoids (GCs) are a class of endocrine disrupting compounds (EDCs) that affect energy metabolism, immune system response, and stress adaption in vertebrate animals [1]. Due to their important physiological functions, various natural and synthetic GCs are highly prescribed drugs for various diseases [1]. It was reported that the consumption of the total prescribed GCs was 9- and 14-fold higher than estrogens and androgens, respectively [2]. The latest research shows that GCs are ubiquitous in environmental water bodies and may have reached a level that negatively affects humans, mammals, and aquatic organisms [3,4]. Research published in *Nature* in 2011 reported that the concentration of the GCs (dexamethasone) in a river in France reached 10 μg/L [5]. Moreover, since GCs are widely being used in patients with the 2019 novel coronavirus (COVID-19) [6], the situation of GC pollution would worsen. Considering the large consumption, ubiquity in environmental water bodies, and potentially negative impacts of GCs, it is urgent to investigate the behavior of GCs in water treatment systems to control GC pollution. However, compared with estrogens, research focused on GC removal is lacking [2].

The limited studies focusing on the removal of GCs in sewage treatment plants (STPs) have shown that traditional biological treatment processes involved in STPs are not fully effective at eliminating GCs [7]. In contrast, the concentrations of some GCs in the final effluent of STPs are even larger than those in influent since their conjugates are hydrolyzed during STP biological treatment [8]. Therefore, conventional STPs should be upgraded with more efficient technologies such as tertiary treatments to provide enough protection for the potential risks related to GCs.

Conventional tertiary treatments are typically involved with chlorination, ozonation ($O_3$), ultraviolet (UV) irradiation, or their combination [9]. Previous studies found that UVC-based treatments and ozonation are two promising ways of controlling GC pollution [2,10,11]. However, as the most commonly used wavelengths, UVC irradiation (emitting practically monochromatic radiation with a maximum at 254 nm) has several drawbacks unsolved, such as the high equipment investment, high operating cost, difficulty to maintain, and potential safety hazards [12]. Recent research has been trying to overcome these limitations by using solar-UVA (emission ranging from 315 to 400 nm, with a maximum at 365 nm) instead of UVC [7,13,14]. Nunes et al. (2020) reported a nonylphenol polyethoxylate (NPEO) degradation efficiency of 96% with the UVA/persulfate process [15]. Wang and Wang (2017) found that chlorophenol could be completely degraded by UVA/persulfate treatment for 1.5 h [16]. Therefore, UVA-based treatments may be good alternatives for UVC-based treatments in removing GCs, which are still waiting to be tested. As chlorine is the most commonly used disinfectant for water and wastewater [17], the combined performance of solar-UVA and chlorine (UVA/chlorination) on GC removal was evaluated in this study. During UVA/chlorination, free active radicals, such as ●OH, Cl●, and $Cl_2^-$●, could be generated by free chlorine (HOCl and $OCl^-$) photolysis as shown in Equations (1)–(8) [2,17], thus accelerating GC degradation. Moreover, as ozonation is also a promising way to control GCs and ozone is also an alternative disinfectant to chlorine [18], the performance of ozonation on GC removal was verified in this study to draw a comparison with that of UVA/chlorination. To our knowledge, scarce comparison analyses of UVA/chlorination and ozonation for GC removal have been carried out, and the GC degradation intermediates and toxicity changes during UVA/chlorination and ozonation are still unknown.

$$HClO + hv \rightarrow \cdot OH + Cl\cdot \tag{1}$$

$$ClO^- + hv \rightarrow \cdot O^- + Cl\cdot \tag{2}$$

$$\cdot OH + OCl^- \rightarrow \cdot OCl + OH^- \tag{3}$$

$$HOCl + \cdot Cl \rightarrow \cdot OCl + Cl^- + H^+ \tag{4}$$

$$OCl^- + \cdot Cl \rightarrow \cdot OCl + Cl^- \tag{5}$$

$$\cdot OH + Cl^- \leftrightarrow ClOH\cdot^- \tag{6}$$

$$ClOH\cdot^- + H^+ \leftrightarrow Cl^- + H_2O \tag{7}$$

$$Cl\cdot + Cl^- \rightarrow Cl_2^-\cdot \tag{8}$$

Among all the GCs, fluocinolone acetonide (FA), triamcinolone acetonide (TA), and clobetasol propionate (CP) deserve particular attention because of their environmental relevance, frequent human exposure, and high environmental toxicity [2]. All three have been found in streams throughout the United States [19], Germany [20], Italy [21], Netherlands [4], Switzerland [22], Czech Republic [23], and Hungary [23] as shown in Table S1 in Supplementary Materials (SI). Their molecular structures are shown in Figure 1. Therefore, the main objectives of this study were to (1) investigate the removal efficiencies of FA, TA, and CP by UVA/chlorination and ozonation under a variety of operation conditions, (2) propose the degradation pathways of FA, TA, and CP during UVA/chlorination and

ozonation, and (3) evaluate the acute toxicity changes of GCs during UVA/chlorination and ozonation. Corresponding results can provide references for GC pollution control.

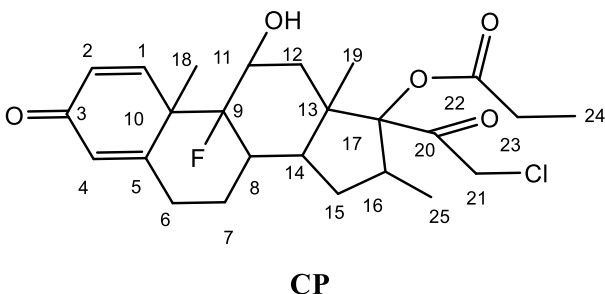

**Figure 1.** The molecular structures and atom label of fluocinolone acetonide (FA), triamcinolone acetonide (TA), and clobetasol propionate (CP).

## 2. Materials and Methods

### 2.1. Chemicals and Reagents

FA (99%), TA (99%), and CP (99%) standards were obtained from Sigma-Aldrich (St. Louis, MO, USA). Stock GC solutions were prepared in acetonitrile at 4 °C in brown bottles. Sodium hypochlorite (NaClO, containing 10% free chlorine as $Cl_2$), high-performance liquid chromatography (HPLC)-grade methanol, and acetonitrile were purchased from Sigma-Aldrich (St. Louis, MO, USA). Other analytically pure chemicals were bought from Sinopharm Group Chemical Reagent Co. Ltd. (Shanghai, China). All solutions were prepared using ultrapure water collected from a Milli-Q system (EMD Millipore, Billerica, MA, USA). Fresh stock solutions of free chlorine and $Na_2SO_3$ were made every time for the experiments. High-pure oxygen ($O_2$) was stored in a high-pressure gas cylinder and supplied by Shanghai Canghai Industrial Gas Co., Ltd. (Shanghai, China).

### 2.2. Experimental Apparatus and Procedures

In the UVA/chlorination system, the reaction device mainly consisted of a 200 mL cylindrical quartz photo-degradation reactor (60 cm (inner) diameter × 11 cm deep) with

a temperature-controlled circulating water interlayer, a collimated beam apparatus using a 200 W short-arc xenon lamp (Beijing Zhongjiao Jinyuan Technology Co., Ltd., Beijing, China.) with an emission wavelength of 315–400 nm, a constant current power supply, and a magnetic stirrer (Shanghai Shenhui Instrument Co., Ltd., Shanghai, China). Typically, 70 mL of the GC aqueous solution was introduced into the photo-degradation reactor with the lamp and stirrer turned on. Then the required amounts of chlorine were immediately spiked into the reactor to start the UVA/chlorination. Samples were withdrawn at specific time intervals, quenched with 15 mM $Na_2SO_3$ at a molar ratio of sulfite/chlorine of 1.5:1, and filtered through a 0.45 μm filter membrane for GC or GC degradation intermediate analysis. The light intensity was measured by a radiometer (Beijing Zhongjiao Jinyuan Technology Co., Ltd.). The temperature was kept at $25 \pm 0.1$ °C. The initial GC concentration was 0.05 mM considering the determination of degradation intermediates. The effects of chlorine dosage (0, 2.5, 5, 10, and 20 mg/L) and light intensity (180, 220, 250, and 280 mW·cm$^{-2}$) on GC removal were evaluated during UVA/chlorination.

All ozonation experiments were performed in a 250-mL-capacity batch bubble column (25 cm height, 6 cm diameter) made of glass and agitated by a magnetic bar. Ozone was produced by a Fischer OZ 500 model ozone generator (Shanghai Zhuoyi Electronics Co., Ltd., Shanghai, China, 250 W) from dry and pure oxygen used as the feed gas. The ozone-containing gas was introduced from the reactor bottom through a sintered glass plate diffuser into the ultrapure water for at least 15 min. The dissolved ozone concentration was detected using a B&C CL7635.010 dissolved ozone online detector (Italy B&C Co., Rome, Italy). The initial pH of the solution was adjusted to the desired value by employing 1 M NaOH or $H_2SO_4$ solutions. After the ozone gas was bubbling for at least 15 min, the desired volume of the stock GC solution was introduced into the reactor and the reaction was started. Samples (0.5 mL) were withdrawn at specific time intervals, quenched, and filtered through a 0.45 μm filter membrane for GC or GC degradation intermediate analysis. The excess ozone leaving the reactor was destroyed by two sequential 20% KI traps incorporated into the reactor setup. The initial concentration of GC was 0.05 mM. The effects of the ozone dosage (19, 31, 38, 59, and 67 mg/L) and initial pH (4.0, 5.0, 6.2, and 7.7) were conducted to probe the performance of the ozonation process. All ozonation experiments were conducted at room temperature.

### 2.3. GC Analysis

For the detection of GCs, we followed the methods described in our previous studies [10,24]. Ultra-high-performance liquid chromatography (UHPLC) (Dionex UltiMate 3000, New York, NY, USA) equipped with a variable wavelength detector was used to analyze the samples at a wavelength of 240 nm. A reversed-phase C-18 column (4.6 mm × 250 mm, 5 μm, Agilent, New York, NY, USA) was used to analyze samples at a temperature of 40 °C. The injection volume was 20 μL and the flow rate was 1 mL min$^{-1}$. The separation was performed under gradient elution conditions using (A) acetonitrile and (B) water. The solvent program was as follows: Initial conditions 72% B kept isocratic for 3 min, linearly reduced to 40% B over 1 min, linearly decreased to 30% B over 4 min, linearly decreased to 0% over 0.5 min, and kept isocratic for at final 1.5 min. The retention time of FA, TA, and CP was approximately 6.2, 7.6, and 8.7 min, respectively. Under these experimental conditions, the quantification limits of FA, TA, and CP were 1.82, 1.71, and 1.47 μM, respectively. The recoveries for FA, TA, and CP were in the range of 85–91%, 85–94%, and 94–102%, respectively. The GC removal efficiency was calculated according to Equation (9) below:

$$Removal\ Efficiency\ (\%) = \frac{C_0 - C_t}{C_0} \times 100\% \tag{9}$$

where $C_0$ and $C_t$ are the initial and final concentrations of GCs, respectively.

### 2.4. Identification of GC Degradation Intermediates

The identification of GC intermediates was achieved using an Agilent 1290 UHPLC coupled with a high-resolution Quadrupole Time-of-Flight (QTOF) mass spectrometer (Agilent 6540 QTOF, New York, NY, USA). Detailed information was described in our previous studies [10,24]. The UHPLC system was employed for separation. The analytical column was an Agilent Zorbax Extend-C18 column (2.1 mm × 100 mm, 1.8 μm). The injection volume was 20 μL and the flow rate was 0.3 mL min$^{-1}$. The samples were separated with a gradient of (A) acetonitrile and (B) water, both acidified with 0.1% acetic acid. The solvent program was as follows: Initial conditions of 98% B kept isocratic for 2 min, then linearly reduced to 2% B over 5 min, then kept isocratic for 2 min. The QTOF mass spectrometer was used under negative electrospray ionization in the TOF mass scan mode. High-resolution mass spectra were collected from 50 to 1700 m/z. The MS parameters were specially optimized as follows: Nebulizer gas (60 psi), capillary voltage (3000 V), and fragmentor (160 V).

### 2.5. Evaluation of the Acute Toxicity

Toxicity changes of FA, TA, and CP after UVA/chlorination and ozonation treatments were evaluated by a bacteria inhibition test using the bioluminescent bacterium *Aliivibrio fischeri* [25]. For UVA/chlorination, samples were taken at 7 h with chlorine dosage of 5 mg/L and light intensity of 280 mW·cm$^{-2}$. For ozonation, samples were taken at 15 min with an ozone dosage of 19 mg/L and an initial pH of 4. Blank samples were prepared simultaneously without any treatment. The bacterial inhibition test was performed via 96-well micro-plate assays. Firstly, the bacteria were incubated in a 5 mL culture medium at 22 °C for 12 h (logarithm phase), and then diluted until the light intensity reached 20,000. The diluted bacteria were added to the 96-well plates that contained the test solutions at different treatment times, and incubated at 22 °C for 15 min. The bioluminescence was determined by Mithras LB 940 Multimode Microplate Reader (Berthold). ($n = 3$ for control and test groups) [25]. The inhibition rate ($I_\%$) of aqueous solution samples on the luminous intensity of photobacterium was calculated according to Equation (10) below:

$$I\ (\%) \ = \frac{I_0 - I_1}{I_0} \times 100\% \tag{10}$$

where $I_1$ is the luminosity of the pollutant sample and $I_0$ is the control luminosity.

### 3. Results and Discussion

#### 3.1. Performance Comparison of UVA/Chlorination and Ozonation on GC Removal

3.1.1. UVA/Chlorination

As shown in Figure 2, UVA/chlorination was effective in GC removal. The decomposition of FA, TA, and CP followed the pseudo-first-order kinetic model during UVA/chlorination (Table S2-1). The UVA light intensity (180–280 mW·cm$^{-2}$) showed more obvious effects on GC removal (Figure 2a–d) than chlorine dosage (0–20 mg/L) (Figure 2e–h), indicating that the dominant degradation method of GCs during UVA/chlorination was photolysis.

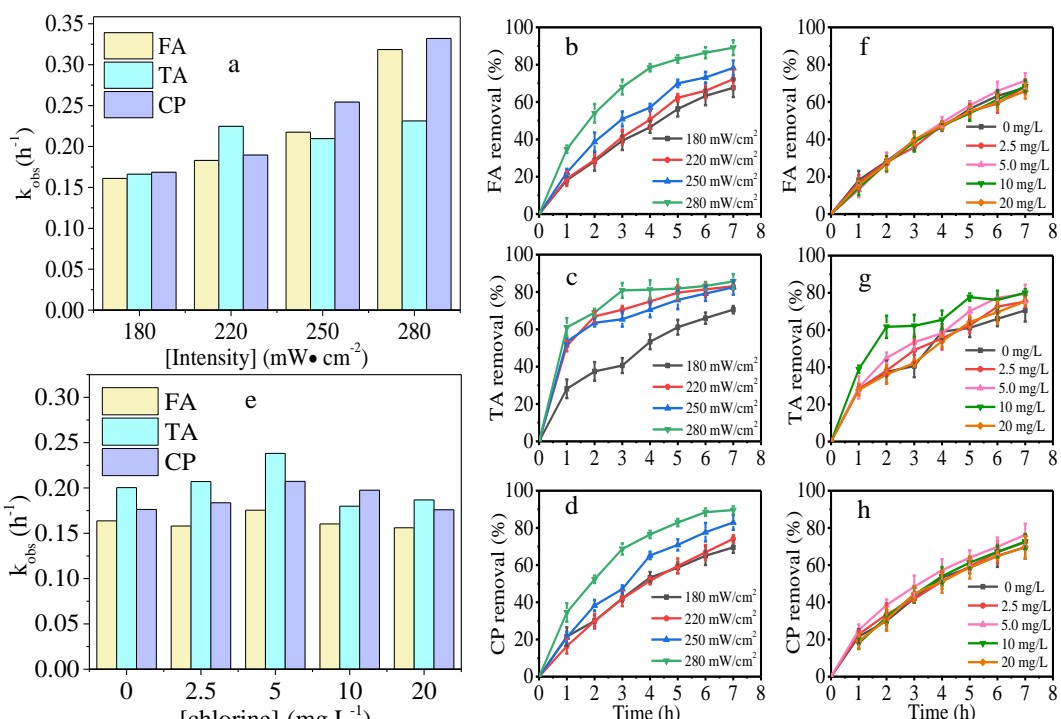

**Figure 2.** Effects of UVA light intensity (**a**) and chlorine dosage (**e**) on rate constant of FA, TA, and CP during UVA/chlorination; effects of light intensity on FA (**b**), TA (**c**), and CP (**d**) removal (chlorine dosage = 5 mg/L); effects of chlorine dosage on FA (**f**), TA (**g**), and CP (**h**) removal (light intensity = 180 mW·cm$^{-2}$).

With the increase in UVA light intensity, the degradation efficiencies and rate constants of FA, TA, and CP were notably enhanced (Figure 2a–d). The removal efficiency of FA, TA, and CP increased from 68% (0.161 h$^{-1}$) to 89% (0.319 h$^{-1}$), 71% (0.166 h$^{-1}$) to 86% (0.231 h$^{-1}$), and 70% (0.169 h$^{-1}$) to 90% (0.332 h$^{-1}$), respectively, while the UVA light intensity increased from 180 to 280 mW·cm$^{-2}$. Similar results were also reported for 2-methylisoborneol and geosmin degradation during UVC irradiation [26]. Under higher UVA radiation intensity, the rearrangement reaction of cross-conjugated dienone or the reaction at ketone of GCs could be dramatically enhanced, thus promoting the GC decomposition [13]. Besides, the synergistic effects of UVA and chlorine became more significant at a higher light intensity because of the acceleration of reactive species generation [27–30].

As shown in Figure 2e–h, with the increase in chlorine dosage from 0 to 5 mg/L at 180 mW·cm$^{-2}$, the rate constants of FA, TA, and CP increased slightly from 0.161, 0.168, and 0.169 h$^{-1}$ to 0.182, 0.223, and 0.177 h$^{-1}$, respectively (Table S2-2). However, with a further increase in the chlorine dosage from 5 to 20 mg/L, the rate constants of FA, TA, and CP decreased (Figure 2e). At a chlorine dosage from 0 to 5 mg/L, the increasing chlorine dosage would favorably produce more free radicals, thus accelerating GC degradation [27]. However, at a chlorine dosage beyond 5 mg/L, the excess free chlorine (such as HOCl and ClO$^-$) would capture ·OH as shown in Equations (11) and (12), thus suppressing the GC removal [31].

$$HOCl + \cdot OH \rightarrow \cdot OCl + H_2O \tag{11}$$

$$OCl^- + \cdot OH \rightarrow \cdot OCl + OH^- \tag{12}$$

Therefore, an optimum chlorine dosage of 5 mg/L was recommended for UVA/chlorination for GC removal. At a chlorine dosage of 5 mg/L and UVA light intensity of 280 mW·cm$^{-2}$, the removal efficiency of FA, TA, and CP during UVA/chlorination could reach 89% (0.319 h$^{-1}$), 86% (0.231 h$^{-1}$), and 90% (0.332 h$^{-1}$) after 7 h, respectively.

### 3.1.2. Ozonation

As shown in Figure 3, GCs could be effectively degraded during ozonation [2]. The degradation processes were fitted with a pseudo-first-order kinetic model (Table S3-1). The GC removal efficiency increased with the increase in the $O_3$ dosage from 19 to 67 mg/L. At an $O_3$ dosage of 19 mg/L, the removal efficiencies of FA, TA, and CP at 15 min were 53, 81, and 85%, respectively, while at an $O_3$ dosage of 67 mg/L, their removal efficiencies increased to over 95%. Correspondingly, the rate constants of FA, TA, and CP increased obviously from 0.068, 0.15, and 0.19 $min^{-1}$ to 0.25, 0.37, and 0.38 $min^{-1}$, respectively (Figure 3a–d). Increasing the ozone equilibrium concentration aggravated the gas–liquid interface agitation, which increased the mass transfer rate of $O_3$ from the gas phase to the liquid phase [32], thus accelerating the GC removal. In addition, the differences in the degradation rates at different GCs may be attributed to the different oxidation states (O/C ratio) in FA, TA, and CP molecules. $O_3$ preferentially reacted with molecules with a low O/C ratio [33]. Therefore, CP was more susceptible to $O_3$ molecules with a lower O/C ratio than FA and TA, leading to the highest degradation rate of CP during ozonation among the three GCs.

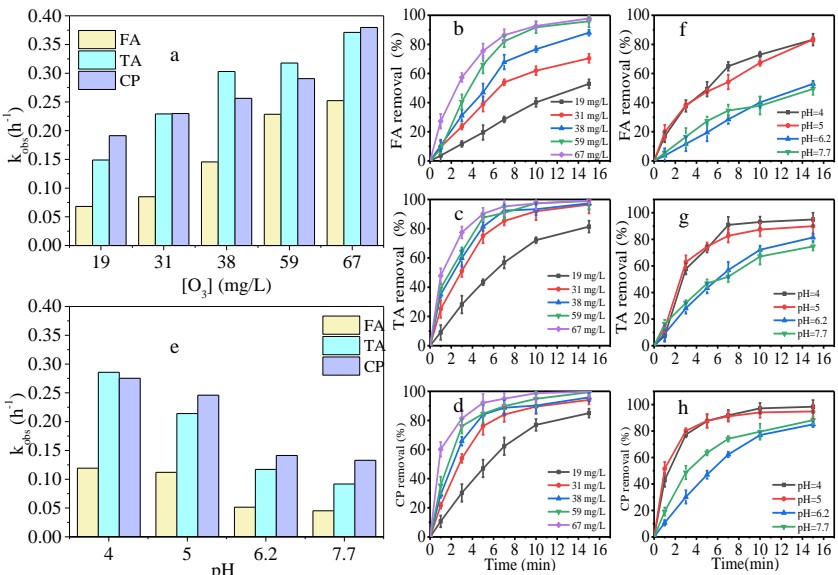

**Figure 3.** Effects of $O_3$ dosage (**a**) and initial pH (**e**) on rate constant of FA, TA, and CP during ozonation; effects of $O_3$ dosage on FA (**b**), TA (**c**), and CP (**d**) removal (initial pH = 6.2); effects of initial pH on FA (**f**), TA (**g**), and CP (**h**) removal (ozone dosage = 19 mg/L).

The effects of the initial pH on GC degradation were also investigated. As shown in Figure 3e–h, the degradation efficiency and rate constant of FA, TA, and CP decreased with the increase in the initial pH value from 4.0 to 7.7 at an $O_3$ dosage of 19 mg/L. The pH value could affect the ozonation kinetic rate and $O_3$ decomposition [34]. $O_3$, as a powerful oxidizing agent (E ($O_3/O_2$) = 2.07 V) [35], could react directly with organic substrates as an electrophilic reagent under acidic conditions [36]. Thus, during the ozonation of GCs under acidic conditions, the $O_3$ molecules could destroy the conjugated π-electron and ring system of GC molecules, leading to GC decomposition [1,37]. However, as the pH values increased, the $O_3$ molecules were easily decomposed and converted to free radicals such as •OH [34]. Although the $O_3$ molecules and •OH radicals could both be favorable to GC decomposition according to our previous study [10], the generation yield of •OH (15–30% [38]) from $O_3$ decomposition could not make up for ozone depletion. Therefore, higher GC removal rates were observed under acidic conditions rather than under alkaline conditions (Figure 3e).

In this study, at an ozone dosage of 19 mg/L and an initial pH of 4, the removal efficiency of FA, TA, and CP during ozonation at 15 min was 90% (0.119 $min^{-1}$), 96%

(0.286 min$^{-1}$), and 98% (0.275 min$^{-1}$), respectively (Table S3-2). In general, compared with UVA/chlorination, GC removal by ozonation treatment is more effective from the viewpoint of decomposition efficiency.

### 3.2. Degradation Pathways of GCs during UVA/Chlorination and Ozonation

Though UVA/chlorination and ozonation can remove GCs from aqueous solutions, the generation of toxic intermediates should be considered [39]. In this study, identification of the GC degradation intermediates during UVA/chlorination and ozonation was involved using the UHPLC-QTOF system [40]. The system provided exact m/z values relating to the chemical formulas as shown in Tables S4 and S5. The experimental m/z ratios had low error (±2.98 ppm), indicating strong confidence in the assignments. The extracted ion chromatogram of the structures and the corresponding mass spectra of the molecular and fragmented ions are displayed in Figure S1. Based on the information of the obtained intermediates, the degradation pathways of FA, TA, and CP during UVA/chlorination and ozonation were proposed.

The proposed structures of FA degradation intermediates during UVA/chlorination and ozonation are illustrated in Figure 4. As shown in Figure 4, chlorine usually undergoes an electrophilic substitution reaction with FA to form a series of chloro-products such as FA-P$_{486}$ [41]. Besides, the free chlorine could also directly oxidize alcohol and ketone groups in the FA structure forming intermediates such as FA-P$_{502}$ and FA-P$_{504}$ [41]. In comparison, the ozone cycloaddition reaction on unsaturated bonds [42] of FA generated ozonides during ozonation. The unstable ozonides would crack to cause ring opening, and further oxidize to keto acids to produce intermediate such as FA-P$_{434}$ [43–45]. In addition, the original FA dioxolane structures were all ring-opened during UVA/chlorination through ●OH-attacking, leading to intermediates such as FA-P$_{394}$, FA-P$_{432a}$, FA-P$_{450a}$, and FA-P$_{448a}$ [10]. In comparison, removing -CH$_2$/-C$_2$H$_4$ from dioxolane structures of FA occurred during ozonation, forming intermediates such as FA-P$_{438}$.

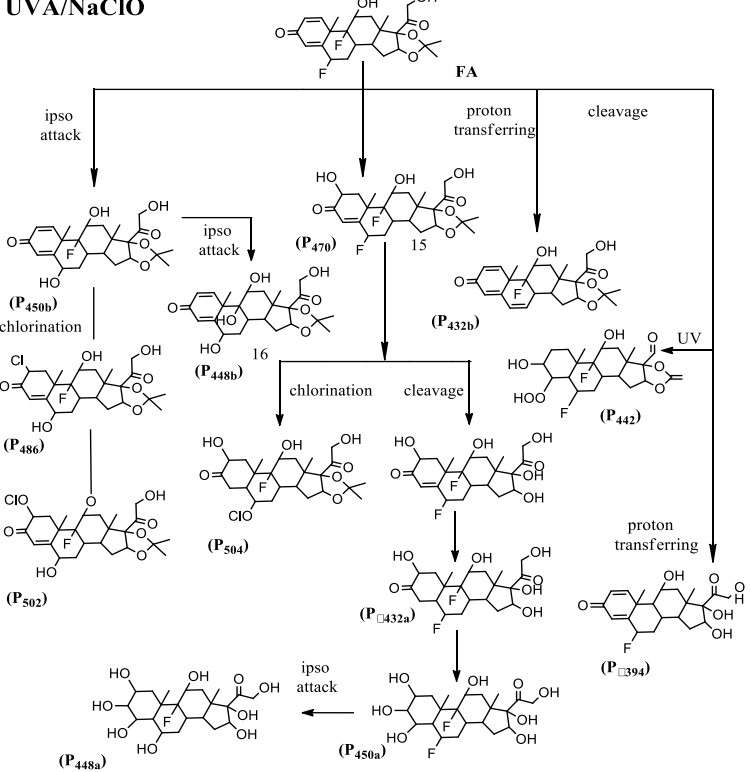

**Figure 4.** *Cont.*

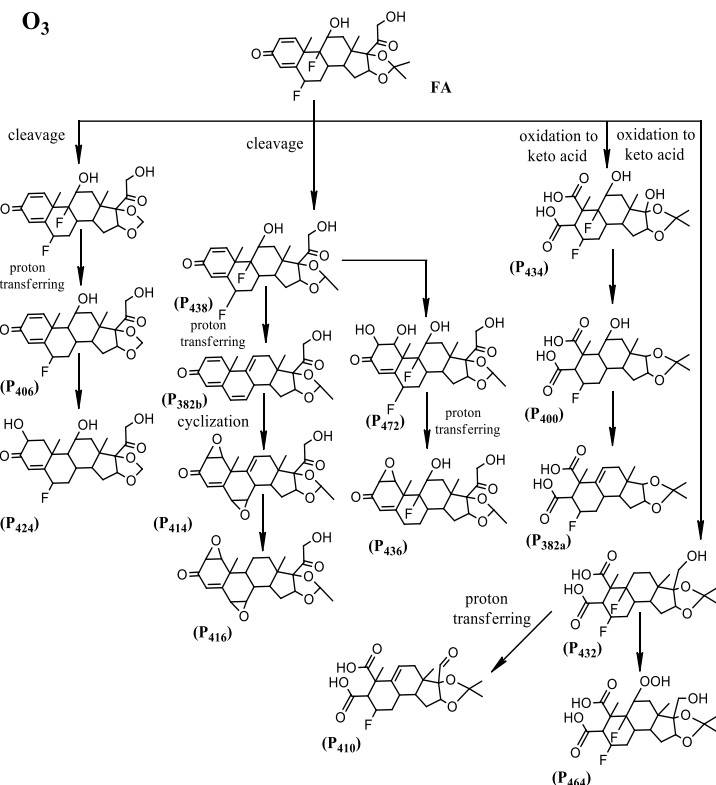

**Figure 4.** Degradation intermediates and reaction pathways of FA during UVA/chlorination (chlorine dosage = 5 mg/L, light intensity = 280 mW·cm$^{-2}$) and ozonation (ozone dosage = 19 mg/L, initial pH = 4).

Reactions with •OH radicals were both observed during UVA/chlorination and ozonation (Figure 4). During UVA/chlorination, •OH radicals could attack the cyclohexenone double bond and generate intermediates such as FA-P$_{470}$. Similarly, cyclohexenone double bond epoxidation (FA-P$_{414}$ and FA-P$_{416}$) and the dehydration of hydroxyl groups (FA-P$_{436}$) occurred during the ozonation process (Figure 4). The intramolecular transfer of protons that removed the F atom was observed during UVA/chlorination, forming the intermediate FA-P$_{432}$ (Figure 4). Similarly, intermediates such as FA-P$_{400}$, FA-P$_{382}$, FA-P$_{406}$, and FA-P$_{410}$ were detected via the intramolecular transfer of protons during ozonation [10,46].

The proposed structures of TA and CP degradation intermediates during UVA/chlorination and ozonation are illustrated in Figures 5 and 6, respectively. Similar to FA, during the UVA/chlorination process, the transfer of intramolecular protons and Cl•/•OCl reactions [10] formed intermediates such as TA-P$_{365}$, TA-P$_{476}$, and TA-P$_{506}$ by removing the F atoms. In addition, the Cl• and •OCl radicals would replace the F atom to produce intermediates such as CP-P$_{512}$ and CP-P$_{482}$ during UVA/chlorination. The photolysis products of TA and CP derived from side-chain cleavage [13] were also detected such as FA-P$_{442}$, TA-P$_{465}$, and CP-P$_{464}$. During ozonation, •OH radicals could attack and oxidize the TA dioxolane structure to open the ring, forming the intermediate TA-P$_{484}$ [10]. Besides, •OH could also attack the ester bonds of the CP molecular and hydrolyze it to alcohols and acids [47], forming the intermediate CP-P$_{444}$.

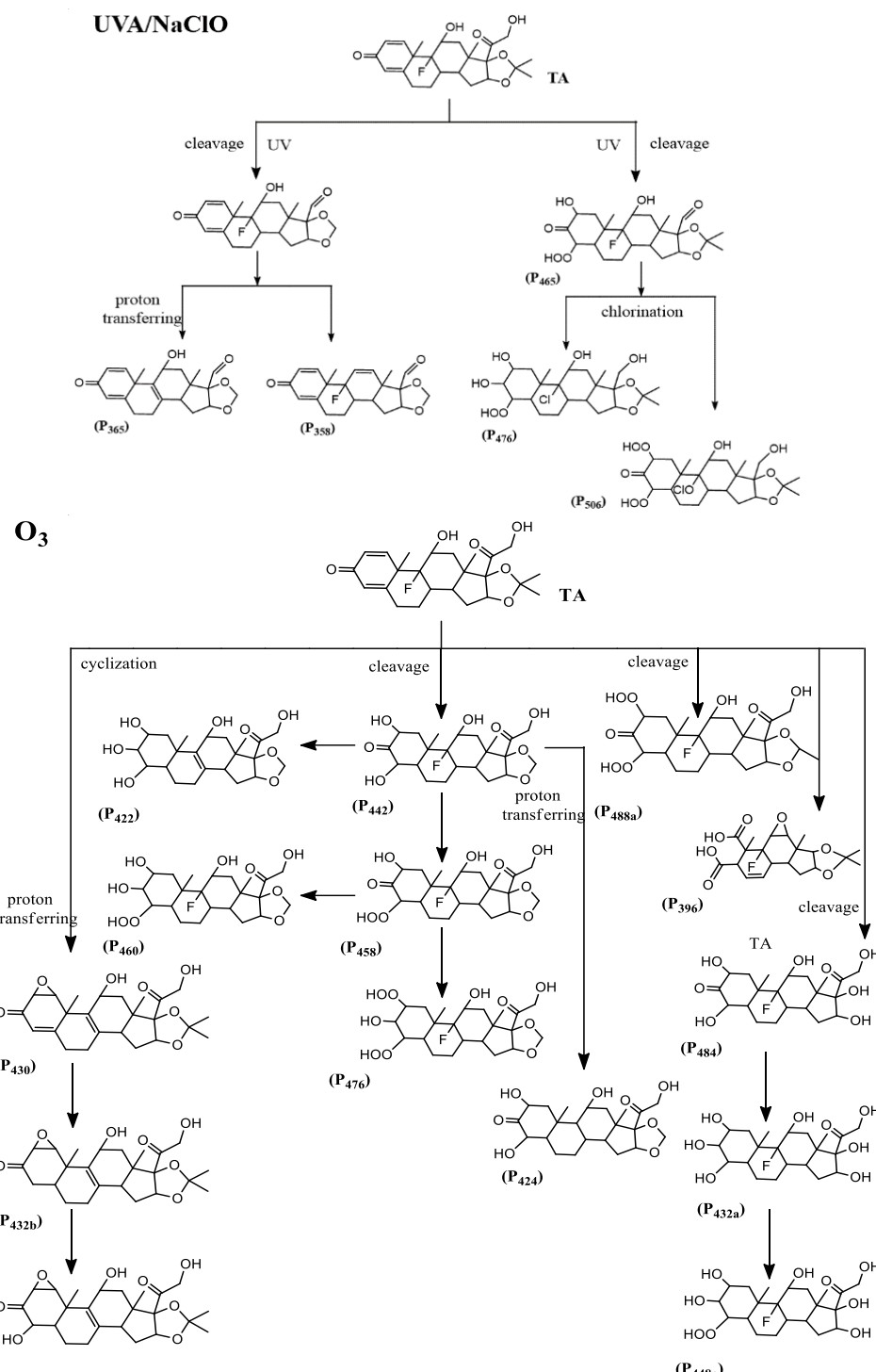

**Figure 5.** Degradation intermediates and reaction pathways of TA during UVA/chlorination (chlorine dosage = 5 mg/L, light intensity = 280 mW·cm$^{-2}$) and ozonation (ozone dosage = 19 mg/L, initial pH = 4).

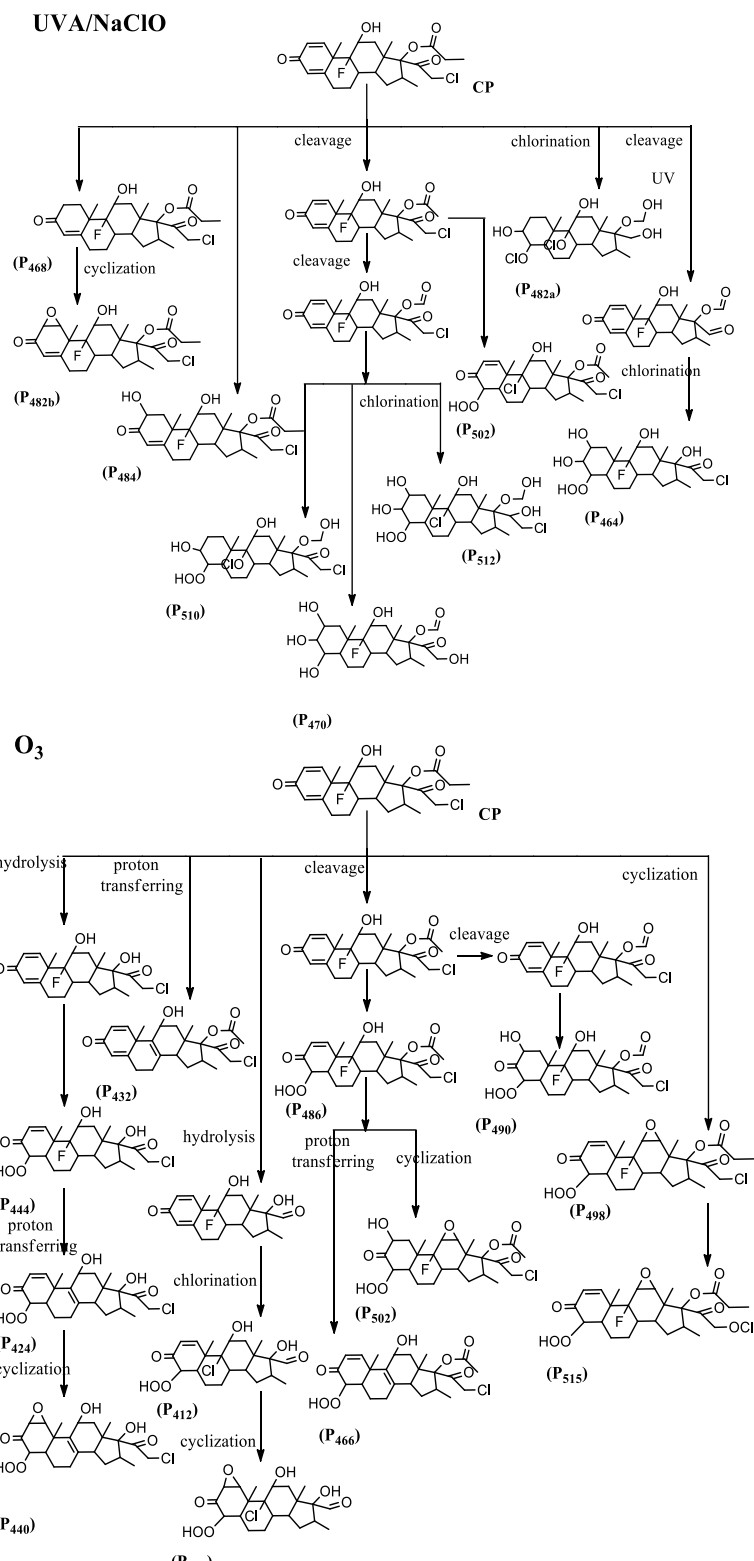

**Figure 6.** Degradation intermediates and reaction pathways of CP during UVA/chlorination (chlorine dosage = 5 mg/L, light intensity = 280 mW·cm$^{-2}$) and ozonation (ozone dosage = 19 mg/L, initial pH = 4).

To summarize, during UVA/chlorination, the degradation pathways of GCs included •OH/Cl• replacing halogen F atoms, side-chain cleavage under UVA irradiation, and •OH addition and ester hydrolysis [46,47]. In comparison, during ozonation, the degradation

pathways of GCs included ozone molecules attacking ring-opening cracking, free radical (•OH and •OH$_2$) bombardment, epoxidation, proton transfer, and dehydration [43–45]. Different from chlorine, which was more likely to attack alcohol and ketone groups during UVA/chlorination, O$_3$ molecules were more likely to attack carbon–carbon double bonds during ozonation [42,48]. Therefore, the benzene ring was cleaved during ozonation, which could convert large molecules to small molecules and oxidize GCs more thoroughly.

### 3.3. Acute Toxicity Change of GCs during UVA/Chlorination and Ozonation

Since new intermediates were generated during UVA/chlorination and ozonation as described in Section 3.2, the toxicity changes of FA, TA, and CP during UVA/chlorination and ozonation were detected by the bacteria inhibition test. As shown in Figure 7, before UVA/chlorination and ozonation, the inhibition rate of FA, TA, and CP was 32, 25, and 37%, respectively. However, after UVA/chlorination treatment for 7 h, the inhibition rate of TA solution increased to 38%, while the inhibition rate of FA solution reduced to 26% and the inhibition rate of CP remained unchanged. In comparison, after ozonation treatment for 15 min, the overall inhibition rate of FA and CP decreased to 27% and 26%, respectively, while the inhibition rate of TA remained unchanged (25%) (Figure 7). Similar results were also reported by Wu et al. [49] who found that the inhibition of *Vibrio fischeri* significantly increased after UVC/chlorination and chlorination. Yin et al. [11] also reported that some intermediates of prednisolone by UVC/chlorine showed higher toxicity than themselves based on the toxicological tests.

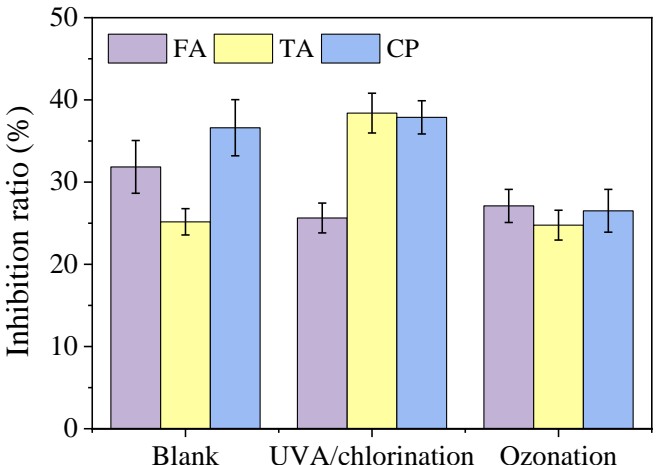

**Figure 7.** Toxicity changes of FA, TA, and CP after UVA/chlorination (chlorine dosage = 5 mg/L, light intensity = 280 mW·cm$^{-2}$, treatment time = 7 h) and ozonation (ozone dosage = 19 mg/L, initial pH = 4, treatment time = 15 min).

Chlorinated wastewater could result in the formation of toxic disinfection by-products (DBPs) [50]. According to Tables S4 and S5, chlorinated intermediates (FA-P$_{486}$, FA-P$_{502}$, FA-P$_{504}$, TA-P$_{476}$, TA-P$_{506}$, CP-P$_{502}$, CP-P$_{510}$, and CP-P$_{512}$) were generated during UVA/chlorination, which may increase the inhibition rate during UVA/chlorination. However, the removal of F and the elimination of epoxidation of double bonds in GCs during UVA/chlorination may reduce the inhibition effects by decreasing the lipophilicity and metabolic resistance of GCs [24,51], causing a decrease in the inhibition rate of the FA solution during UVA/chlorination (Figure 7). In comparison, more dechlorination intermediates were observed during ozonation than that during UVA/chlorination (Figures 4–6), leading to a lower inhibition rate during ozonation (Figure 7). Therefore, ozonation was more effective in GC degradation and toxicity reduction than UVA/chlorination.

*3.4. Implications*

Although most GCs were removed during ozonation (90, 96, and 98% for FA, TA, and CP, respectively) and UVA/chlorination (89, 86, and 90% for FA, TA, and CP, respectively), their acute toxicities declined slightly (from 32, 25, and 37% for the blank to 27, 25, and 26% after ozonation and 26, 38, and 37% after UVA/chlorination for FA, TA, and CP, respectively). Since the stock GC solutions were prepared in acetonitrile, the decrease in the total organic carbon (TOC) could not reflect the mineralization of the three selected GCs. The residual acute toxicities and various detected intermediates indicated that both ozonation and UVA/chlorination could not mineralize GCs completely. Therefore, subsequent treatments are needed to mineralize the residual GC intermediates and eliminate their toxicities.

Since GCs are widely used in patients with COVID-19 [6], various GCs have been found in the hospital wastewater of many countries at high concentrations [4]. The currently available hospital wastewater treatment methods mainly include traditional biological treatment processes, ozonation, UV irradiation, chlorination, and their combination [52]. As traditional biological treatment processes are not fully effective in eliminating GCs [7] and ozonation and UVA/chlorination could not fully mineralize GCs and eliminate their toxicities (this study), other effective treatment technologies are needed to remove GCs, with more attention focused on intermediates and toxicity changes. A comprehensive evaluation of the intermediates and toxicity changes along water treatment technology is highly recommended to successfully evaluate the removal efficiencies of GCs.

## 4. Conclusions

Both UVA/chlorination and ozonation could remove GCs. Compared with UVA/chlorination, ozonation was more effective in GC degradation. At a chlorine dosage of 5 mg/L and UVA light intensity of 280 mW·cm$^{-2}$, the removal efficiency of FA, TA, and CP during UVA/chlorination reached 89%, 86%, and 90% after 7 h, respectively. In comparison, at an ozone dosage of 19 mg/L and an initial pH of 4, the removal efficiency of FA, TA, and CP during ozonation reached 90%, 96%, and 98% at 15 min, respectively.

During UVA/chlorination, GC pollutants were mainly photodegraded by UVA irradiation, while, during ozonation, GCs were oxidized by ozone molecules. The main reaction processes for GC degradation mainly included H-abstraction, dechlorination, carbon–carbon bond cleavage, and ester hydrolysis during ozonation and UVA/chlorination. Specifically, ozone tended to crack benzene rings of GCs by attacking C=C double bonds while chlorine tended to attack alcohol and ketone groups of GCs. Though GCs were effectively degraded during ozonation and UVA/chlorination, their degradation products could not be completely mineralized, resulting in unabated toxicity. Compared with UVA/chlorination, ozonation was more effective in toxicity reduction. Subsequent treatments are needed to mineralize the residual GC intermediates and eliminate their toxicities.

**Supplementary Materials:** The following supporting information can be downloaded at: https://www.mdpi.com/article/10.3390/w14162493/s1. Table S1: Available data on measured concentrations of FA, TA, and CP in various environmental samples; Table S2-1: Pseudo-first-order kinetic models and corresponding R2 of GCs under different light intensities during UVA/chlorination; Table S2-2: Pseudo-first-order kinetic models and corresponding R2 of GCs under different chlorine dosages during UVA/chlorination; Table S3-1 Pseudo-first-order kinetic models and corresponding R2 of GCs under different O3 dosages during ozonation; Table S3-2 Pseudo-first-order kinetic models and corresponding R2 of GCs under different initial pH values during ozonation; Table S4-1 General information of identified transformation products of FA during UVA/chlorination; Table S4-2 General information of identified transformation products of TA during UVA/chlorination; Table S4-3 General information of identified transformation products of CP during UVA/ chlorination; Table S5-1 General information of identified transformation products of FA during ozonation; Table S5-2 General information of identified transformation products of TA during ozonation; Table S5-3 General information of identified transformation products of CP during ozonation; Figure S1-1 Extracted ion chromatographs and mass spectrometry fragments for intermediates of GCs during UVA/chlorination;

Figure S1-2 Extracted ion chromatographs and mass spectrometry fragments for intermediates of GCs during ozonation.

**Author Contributions:** Conceptualization, J.W.; methodology, A.Z., X.J., Q.W. and S.H.; validation, Q.W. and S.H.; formal analysis, D.Z. and C.W.; data curation, A.Z. and M.L.; writing—original draft preparation, A.Z., Q.W. and X.J.; writing—review and editing, D.Z., C.W. and M.L.; supervision, J.W.; funding acquisition, A.Z. and J.W. All authors have read and agreed to the published version of the manuscript.

**Funding:** This research has been co-financed by the Shanghai Chen-Guang Program [19CG38], the Natural Science Foundation of Shanghai (22ZR1402800), and the National Natural Science Foundation of China (51708096). All financial support is gratefully acknowledged.

**Institutional Review Board Statement:** Not applicable.

**Informed Consent Statement:** Not applicable.

**Data Availability Statement:** The datasets generated during and/or analyzed during the current study are available from the corresponding author on reasonable request.

**Conflicts of Interest:** The authors declare no conflict of interest.

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
