# Peer review of "Assessment of Glucocorticoid Removal by UVA/Chlorination and Ozonation: Performance Comparison in Kinetics, Degradation Pathway, and Toxicity"

_water, doi:10.3390/w14162493_

Round 1
Reviewer 1 Report
This article focuses on a popular topic, design sufficient experiments and performed meaningful research. Methods are not new but used in a meaningful way. Some minor changes could help improve the quality of the paper:
1. The last several sentences of Abstracts and Conclusions are very similar, this could be considered as self-plagiarism. The authors have to rewrite either the abstract or the conclusion.
2. The authors cited a lot of reference in the results and discussion section. This is not good scientific practice. Either discuss the mechanism in the Introduction section, or have a separate section to discuss the mechanism and focus on the degradation pathway. The structure of the paper needs to be changed.
Author Response
Dear Reviewer,
Thank you for your comments concerning our manuscript entitled “Assessment of glucocorticoid removal by UVA/chlorination and ozonation: Performance comparison in kinetics, degradation pathway, and toxicity” (water-1847528). Those comments are all valuable and very helpful for revising and improving our paper, as well as the important guiding significance to our research. We have studied the comments meticulously and have made corrections which we hope to meet with approval. Revised portions are marked in red in the revised manuscript. The main corrections in the paper and the point-by-point responses to the editor’s and reviewers’ comments (comments in blue, our replies in black, and revises in red) are as follows:
REVIEWER #1:
This article focuses on a popular topic, design sufficient experiments and performed meaningful research. Methods are not new but used in a meaningful way. Some minor changes could help improve the quality of the paper:
- The last several sentences of Abstracts and Conclusions are very similar; this could be considered as self-plagiarism. The authors have to rewrite either the abstract or the conclusion.
Reply: We are grateful for the positive comments and suggestions to improve the quality and readability of the manuscript. The conclusion has been rewritten.
Revision: (Rewrited the last paragraph of the Conclusions)
During UVA/chlorination, GC pollutants were mainly photodegraded by UVA irradiation, while during ozonation, GCs were oxidized by ozone molecules. The main reaction processes for GC degradation mainly included H-abstraction, dechlorination, carbon-carbon bond cleavage, and ester hydrolysis during ozonation and UVA/chlorination. Specifically, ozone tended to crack benzene rings of GCs by attacking C=C double bonds while chlorine tended to attack alcohol and ketone groups of GCs. Though GCs were effectively degraded during ozonation and UVA/chlorination, their degradation products could not be completely mineralized, resulting in unabated toxicity. Compared with UVA/chlorination, ozonation was more effective in toxicity reduction. Subsequent treatments are needed to mineralize the residual GC intermediates and eliminate their toxicities.
- The authors cited a lot of reference in the results and discussion section. This is not good scientific practice. Either discuss the mechanism in the Introduction section, or have a separate section to discuss the mechanism and focus on the degradation pathway. The structure of the paper needs to be changed.
Reply: We are grateful for the positive comments and suggestions to improve the quality and readability of the manuscript. We added the discussion of the mechanisms in the “Introduction” section and deleted some references and discussion in the “Results and Discussion” section.
Revision: (Added the discussion of the mechanisms in the “Introduction” section)
As chlorine is the most commonly used disinfectant for water and wastewater [17], the combination performance of solar-UVA and chlorine (UVA/chlorination) on GC removal were evaluated in this study. During UVA/chlorination, free active radicals, such as •OH, Cl•, and Cl2-•, could be generated by free chlorine (HOCl and OCl-) photolysis as shown in Eqs. 1-8 [2, 17], thus accelerating GC degradation.
(1)
(2)
(3)
(4)
(5)
(6)
(7)
(8)
Added reference: X. Kong, J. Jiang, J. Ma, Y. Yang, S. Pang, Comparative investigation of X-ray contrast medium degradation by UV/chlorine and UV/H2O2, Chemosphere. 193 (2018) 655-663, https://doi.org/10.1016/j.chemosphere.2017.11.064.
(Deleted some references and discussion in the “Results and Discussion” section)
However, further increase the chlorine dosage from 5 to 20 mg/L, the rate constants of FA, TA, and CP decreased (Fig. 2e). During UVA/chlorination, free active radicals, such as •OH, Cl•, and Cl2-•, could be generated by free chlorine (HOCl and OCl-) photolysis as shown in Eqs. 3-6 [2, 27, 31].
Deleted reference: T.R. John CC, David WH, Kerry JH, George T, Disinfection/Oxidation by-products, in MWH's Water Treatment: Principles and Design, Third Edition, 2012, Wiley Online Library. p. 1485-1527, https://onlinelibrary.wiley.com/doi/book/1410.1002/9781118131473.

Reviewer 2 Report
The presence of pharmaceuticals and their derivatives in the aquatic environment is a serious problem for the entire ecosystem. Leading reserch centers in the world deal with their elimination from water and sewage. So the peer-reviewed article on the removal of GCs, FP, TA and CP by UVA/Cl2 and O3 fromaqueous solution fits perfectly in this trend. In the opinion of the Reviewer, the Authors of the article correctly formulated the purpose and scope of the research, used appropriate analitical techniques in the research, correctly inerpreted the research results and on the based of them correctly formulated the final conclusions. Due to the research material and method of conducting research, the article is of a cognitive nature, while the results and conclusions presented in it are currently difficult to use in practice, for example for economic reasons. The Reviewer believes that Authors should include in the research:
1/ combination of O3 and UV, i.e. the O3/UV process, which belongs to the AOPs group and is highly effective in removing various types of organic substances of an anthropogenic orgin that are difficult to decompose,
2/ measurement of the degree of mineralization of the oxidized substances depending on the combination of the used oxidants and related technological parameters.
Regardless of the above, the Reviewer believes that the article is a significant contribution to knowledge and therfore the Reviewer recommends it to be published in the journal Water.
Author Response
Comments and Suggestions for Authors
-2
Dear Reviewer,
Thank you for your comments concerning our manuscript entitled “Assessment of glucocorticoid removal by UVA/chlorination and ozonation: Performance comparison in kinetics, degradation pathway, and toxicity” (water-1847528). Those comments are all valuable and very helpful for revising and improving our paper, as well as the important guiding significance to our research. We have studied the comments meticulously and have made corrections which we hope to meet with approval. Revised portions are marked in red in the revised manuscript. The main corrections in the paper and the point-by-point responses to the editor’s and reviewers’ comments (comments in blue, our replies in black, and revises in red) are as follows:
REVIEWER #2:
The presence of pharmaceuticals and their derivatives in the aquatic environment is a serious problem for the entire ecosystem. Leading research centers in the world deal with their elimination from water and sewage. So the peer-reviewed article on the removal of GCs, FP, TA and CP by UVA/Cl2 and O3 from aqueous solution fits perfectly in this trend. In the opinion of the Reviewer, the Authors of the article correctly formulated the purpose and scope of the research, used appropriate analytical techniques in the research, correctly interpreted the research results and on the based of them correctly formulated the final conclusions. Due to the research material and method of conducting research, the article is of a cognitive nature, while the results and conclusions presented in it are currently difficult to use in practice, for example for economic reasons. The Reviewer believes that Authors should include in the research:
1/ combination of O3 and UV, i.e. the O3/UV process, which belongs to the AOPs group and is highly effective in removing various types of organic substances of an anthropogenic orgin that are difficult to decompose,
Reply: We are grateful for the positive comments and suggestions to improve the quality and readability of the manuscript. We plan to study the efficiency of O3/UV on GC removal in the future. As the results in this study indicated that O3 and UV were effective in GC removal, we believe that the combination of ozonation and UV irradiation could have excellent efficiency in GC degradation. Considering that this manuscript already had enough data on GC removal by ozonation and UV/chlorination, the research on the combination of O3 and UV will be reported in the next article.
2/ measurement of the degree of mineralization of the oxidized substances depending on the combination of the used oxidants and related technological parameters.
Reply: We are grateful for the positive comments and suggestions to improve the quality and readability of the manuscript. Due to the low solubility of the target GCs in water, we had to add organic solvent to the reaction solutions to help solubilization of GCs. The organic solvent was acetonitrile which would not be oxidized by the oxidants used in our study. By adding acetonitrile, the initial GC concentration was high enough for us to detect GC intermediates, which was the most important scientific discovery reported in this manuscript. However, due to the addition of organic solvent, it became difficult for us to monitor the mineralization of GCs by measuring total organic carbon concentrations (TOC). Therefore, measurements of the degree of mineralization of the oxidized substances were not conducted in this study. We will explore the mineralization degree of GCs during various AOPs in the future.
Regardless of the above, the Reviewer believes that the article is a significant contribution to knowledge and therefore the Reviewer recommends it to be published in the journal Water.

Reviewer 3 Report
Please cite main recent books and reviews in this area. Also, the free-radical mechanism (reactions 3-8) seems too simplified. Please develop and expand it.
In whole, the manuscript is good and can be published after the changes above.
Author Response
Comments and Suggestions for Authors
-3
Dear Reviewer,
Thank you for your comments concerning our manuscript entitled “Assessment of glucocorticoid removal by UVA/chlorination and ozonation: Performance comparison in kinetics, degradation pathway, and toxicity” (water-1847528). Those comments are all valuable and very helpful for revising and improving our paper, as well as the important guiding significance to our research. We have studied the comments meticulously and have made corrections which we hope to meet with approval. Revised portions are marked in red in the revised manuscript. The main corrections in the paper and the point-by-point responses to the editor’s and reviewers’ comments (comments in blue, our replies in black, and revises in red) are as follows:
REVIEWER #3:
Please cite main recent books and reviews in this area. Also, the free-radical mechanism (reactions 3-8) seems too simplified. Please develop and expand it.
Reply: We are grateful for the positive comments and suggestions to improve the quality and readability of the manuscript. We have cited some books and reviews in this area as follows:
- Wardenier, Z. Liu, A. Nikiforov, S.W.H. Van Hulle, C. Leys, Micropollutant elimination by O3, UV and plasma-based AOPs: An evaluation of treatment and energy costs, Chemosphere. 234 (2019) 715-724, https://doi.org/10.1016/j.chemosphere.2019.06.033;
T.R. John CC, David WH, Kerry JH, George T, Disinfection/Oxidation by-Products, in MWH's Water Treatment: Principles and Design, Third Edition, 2012, Wiley Online Library. p. 1485-1527, https://onlinelibrary.wiley.com/doi/book/1410.1002/9781118131473;
- Asghar, A.A.A. Raman, W.M.A.W. Daud, Advanced oxidation processes for in-situ production of hydrogen peroxide/hydroxyl radical for textile wastewater treatment: a review, J. Clean. Prod. 87 (2015) 826-838, https://doi.org/10.1016/j.jclepro.2014.09.010;
- von Gunten, Ozonation of drinking water: Part I. Oxidation kinetics and product formation, Water Res. 37 (2003) 1443-1467, https://doi.org/10.1016/s0043-1354(02)00457-8;
The free-radical mechanism (reactions 1-8) was developed and expanded in the “Introduction” section.
Revision: (Added the discussion of the mechanisms in the “Introduction” section)
As chlorine is the most commonly used disinfectant for water and wastewater [17], the combination performance of solar-UVA and chlorine (UVA/chlorination) on GC removal were evaluated in this study. During UVA/chlorination, free active radicals, such as •OH, Cl•, and Cl2-•, could be generated by free chlorine (HOCl and OCl-) photolysis as shown in Eqs. 1-8 [2, 17], thus accelerating GC degradation.
(1)
(2)
(3)
(4)
(5)
(6)
(7)
(8)
Added reference: X. Kong, J. Jiang, J. Ma, Y. Yang, S. Pang, Comparative investigation of X-ray contrast medium degradation by UV/chlorine and UV/H2O2, Chemosphere. 193 (2018) 655-663, https://doi.org/10.1016/j.chemosphere.2017.11.064.
In whole, the manuscript is good and can be published after the changes above.
